# Reconstruction of 3D Shower Shape with the Dual-Readout Calorimeter

**Sanghyun Ko** [1,*], **Hwidong Yoo** [2] **and Seungkyu Ha** [2] **on behalf of the IDEA Dual-Readout Group**

1   Department of Physics and Astronomy, Seoul National University, Seoul 08826, Korea
2   Department of Physics, Yonsei University, Seoul 03722, Korea
*   Correspondence: sanghyun.ko@cern.ch

**Abstract:** The dual-readout calorimeter has two channels, Cherenkov and scintillation, that measure the fraction of an electromagnetic (EM) component within a shower by using different responses of each channel to the EM and hadronic component. It can measure the energy of EM and hadronic shower simultaneously—its concept inspired the integrated design for measuring both EM and hadronic showers, which left the task of reconstructing longitudinal shower shapes to the utilization of timing. We explore the possibility of longitudinal shower shape reconstruction using signal processing on silicon photomultiplier timing, and 3D shower shape by combining lateral and longitudinal information. We present a comparison between Monte Carlo (MC) and reconstructed longitudinal shower shapes from the simulation, and the application of 3D shower shapes associated with the dual nature of the calorimeter to identify electrons, hadrons, and hadronic punch-thru or muons.

**Keywords:** calorimetry; particle detectors; photomultipliers; particle identification

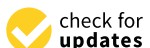



## 1. Introduction

In high-energy physics experiments, the energy of incident particles can be measured through a destructive interaction with absorbers that generates subsequent showers. A hadronic particle can develop not only the hadronic component that mainly consists of charged mesons and protons, but also the EM component from the production and decay of neutral pions. However, the detector's response to the hadronic component is significantly lower than that of the EM component for most calorimeters. Hence, the fluctuation of EM fraction within the shower initiated by hadronic particles limits the equivalent measurement of hadronic and EM components, hindering measuring energy.

The dual-readout calorimeter [1,2] is one of the proposed solutions to counter the fluctuation of EM fraction within the hadronic shower, simultaneously measuring the EM and hadronic components by utilizing different responses of Cherenkov and scintillation channels to relativistic and nonrelativistic particles. Each channel has a distinct response ratio for the hadronic component to the EM component, and the ratio of Cherenkov to scintillation channel (C/S) allows for estimating the EM fraction within the shower.

The ability to measure the energy of EM and hadronic showers simultaneously has led to contemporary designs of the dual-readout calorimeter that have no longitudinal segmentation. However, the longitudinal profile of a hadronic shower carries certain information that may improve the particle identification and energy reconstruction performance of hadronic showers. Studies with 3D-segmented particle flow calorimeters [3] suggest that details of shower shapes can be used for the software compensation technique. For instance, EM parts of the shower are more compact and denser compared to hadronic parts of the shower due to different scales between the radiation length and the nuclear interaction length. Therefore, we try to exploit timing information to reconstruct longitudinal and 3D shower shapes for a dual-readout calorimeter without physical segmentation.

## 2. Simulation Setup

The $4\pi$ projective geometry of the dual-readout calorimeter was implemented in the simulation with DD4hep [4]. Figure 1 illustrates the arrangement of Cherenkov and scintillation fibers with a 3D-printed projective module. A unit module is a trapezoidal tower consisting of a copper absorber with a 2 m length in the longitudinal direction. Inside the tower, scintillation and Cherenkov fibers were inserted in a checkerboard pattern at a 1.5 mm distance between fibers.

In the dual-readout calorimeter, fibers' optical properties determine the timing characteristics. Therefore, detailed descriptions of optical properties are essential. The Cherenkov fiber implemented in the simulation was a Mitsubishi Eska SK-40 clear fiber with a polymethylmethacrylate (PMMA) core and fluorinated polymer cladding. The scintillation fiber was Kuraray SCSF-78, consisting of PMMA cladding and polystyrene-based scintillating core. We emulated the refractive index [5–7], attenuation length [8,9], light yield, emission spectra, and decay time in the detector descriptions [10].

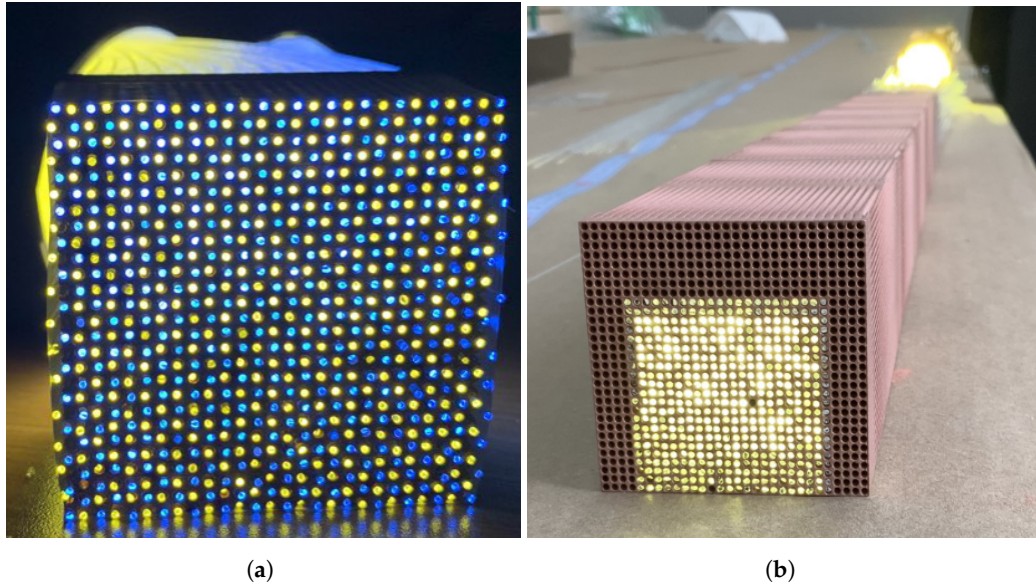

(**a**)　　　　　　　　　　　　　　　　　　　　　　　　　　(**b**)

**Figure 1.** (**a**) Dual-readout calorimeter with Cherenkov fibers (blue light) and scintillation fibers (yellow light). (**b**) Projective geometry of dual-readout calorimeter with a copper absorber where only lit fibers on the rear side had full length that reached the front of the tower.

The above detector descriptions in DD4hep were interfaced to a GEANT4 [11–13] MC simulation. Figure 2 shows optical physics within the fibers simulated with GEANT4, where we can observe the unique behaviors of Cherenkov and scintillation light emission, and the propagation of optical photons via total internal reflection.

Generated optical photons are detected at the rear end of the tower. The collected number of photons, time of arrival, and wavelength information are plugged into silicon photomultiplier (SiPM) emulation software library SimSiPM [14]. This describes the response of SiPMs driven by parameters obtained from either lab measurements or data sheets from manufacturers. In the simulation, the data sheet of Hamamatsu S14160-1310PS SiPM [15] was used to describe SiPM behaviors, including dark count rate, afterpulse, cross-talk, and pulse shape as a function of time. Between the rear ends of scintillation fibers and SiPMs, Kodak Wratten number 9 yellow filters were inserted, which prevents the saturation of SiPMs from the high light yield of the scintillation channel, and absorbs the spatially dependent short-attenuation-length blue light.

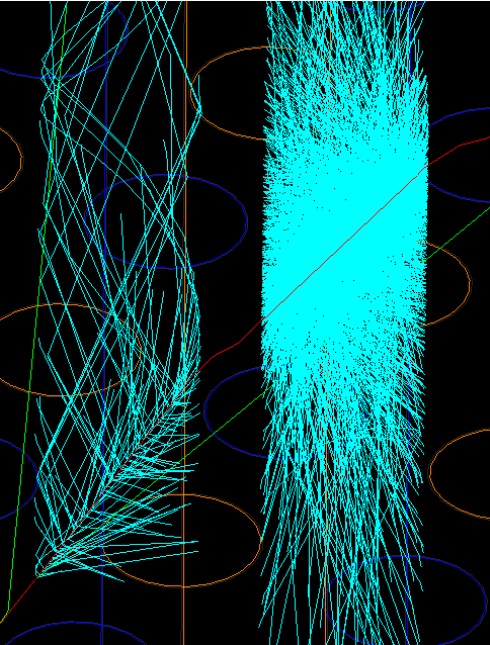

**Figure 2.** Optical physics simulated with GEANT4. The red line from the bottom left to the upper right indicates the electron from the shower fragment. The green line is a low-energy photon from the radiation, and cyan lines are optical photons generated from Cherenkov and scintillation process on the left and right sides, respectively. Blue and orange lines represent a sectional view of Cherenkov and scintillation fibers.

## 3. Longitudinal and 3D Shower Shape Reconstruction

### 3.1. Removing the Exponential Decay Signature

For fiber-sampling calorimeters such as the dual-readout calorimeter, the conventional approach to obtain longitudinal information regarding the shower is to estimate it using the time typically taken from the signal's peak or time of arrival. Setting the impact point and the moment of collision as $\vec{x} = 0$, $t = 0$, the observed time can be expressed as the sum of a high-energy particle's time of flight (ToF) and the propagation time of an optical photon within the fiber with group velocity $v$. SiPM's position $\vec{l}$ can also be described as a vector sum of the flight path of the high-energy particle and the distance that the optical photon propagated.

$$t = \frac{|\vec{x}|}{c} + \frac{|\vec{k}|}{v} \qquad \vec{l} = \vec{x} + \vec{k} \tag{1}$$

Here, we can benefit from the projective geometry to reconstruct the position of energy deposits by approximating that the three vectors are almost parallel.

$$|\vec{k}| = \frac{t - |\vec{l}|/c}{1/v - 1/c} \qquad \vec{x} = \vec{l} - \frac{t - |\vec{l}|/c}{1/v - 1/c} \frac{\vec{k}}{|\vec{k}|} \tag{2}$$

However, using only the time of a peak or arrival yields only a single number per shower, eventually ignoring details aside from the depth of the shower maximum or the tail. Therefore, understanding the full details of longitudinal shower shapes require utilizing the entire timing structure.

Without longitudinal segmentation, the calorimeter solely depends on the timing to reconstruct longitudinal shower shapes. Unfortunately, interpreting an electronic pulse shape into a physical shower shape is very challenging due to the many hidden layers between the two. For instance, a SiPM does not show a narrow pulse from a single photon. Instead, it returns an exponentially decaying pulse with a relatively short rise time compared to the decay time. Moreover, the number of photons follows the exponential decay by the scintillation process even emitted at the same depth.

Fortunately, the common nature of exponential decay allows for us to establish the energy density contributed to the pulse shape we observed from the SiPM by using the classic Fourier transform technique. For example, a pulse shape can be modeled as a convolution of exponential decay with time translation.

$$f(t) = \Theta(t - t_0)e^{-k(t - t_0)} \tag{3}$$

The exponential decay is described as a Lorentzian function in the frequency domain, while time translation becomes an oscillating component. $\Delta t$ is a unit time of the discrete Fourier transform, corresponding to the sampling time of electronics.

$$F(\omega) = \frac{1}{1 - e^{-(k + i\omega)\Delta t}} \tag{4}$$

Provided that the time translation (time of arrival) given by the time window is not too large compared to the decay time of SiPMs, we can roughly interpret the full-width half maximum (FWHM) $\Delta\omega$ as an effective decay time.

$$cosh(k\Delta t) = 2 - cos(\Delta\omega\Delta t/2) \tag{5}$$

Hence, we can remove the decay term while leaving the oscillating component untouched, yielding time translation information solely without the exponential decay.

$$\frac{F(\omega)}{1 - e^{-(k + i\omega)\Delta t}} \to F(\omega) \tag{6}$$

Figure 3 is a signal-processing example simulated with SimSiPM: 1 ns rise time and 6.5 ns decay time based on the Hamamatsu S14160-1310PS data sheet but with higher signal-to-noise (SNR) ratio for the clear demonstration. It represents an analog signal consisting of four pulses—a two-photon equivalent pulse from 13 to 14 ns, the main five-photon equivalent pulse from 15 to 16 ns, followed by two late single photon contributions at 18 and 25 ns. As photons are collected at the rear side of the tower, understanding the shape of early contributions is essential for reconstructing the tail part of the shower. However, it is challenging to discriminate them from the primary pulse, as shown in Figure 3a. Figure 3c demonstrates that signal processing significantly reduces decay structures for each pulse; hence, we can recognize individual contributions within the signal.

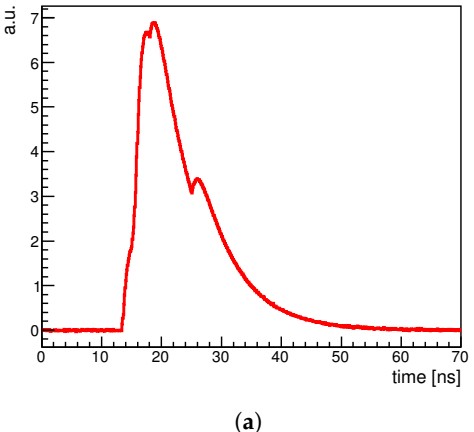

(a)

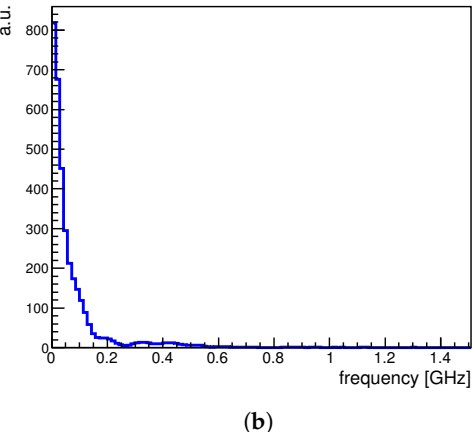

(b)

**Figure 3.** *Cont.*

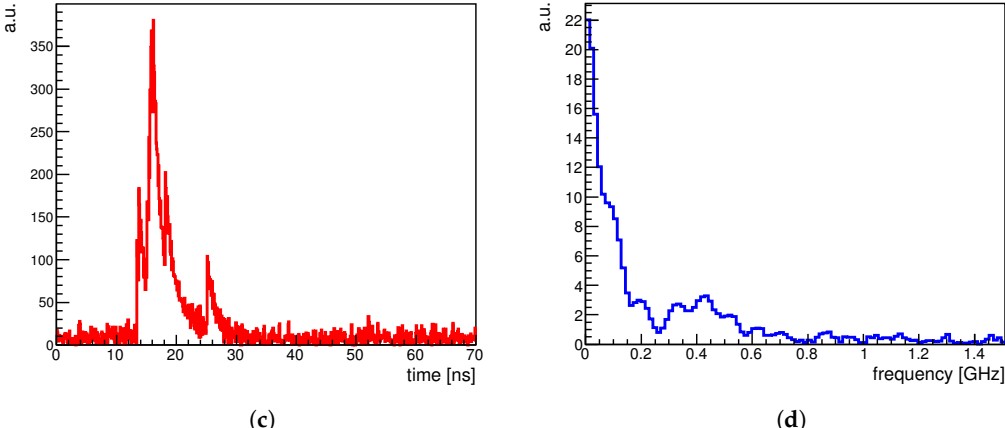

(**c**)                                           (**d**)

**Figure 3.** An example of a simulated signal in (**a**) time and (**b**) frequency domains. Signal after decay removal in the (**c**) time and (**d**) frequency domains, respectively.

### 3.2. Mitigating Modal Dispersion of Optical Fibers

However, the signal pulse shape is still far from the physical shower shape even after removing the exponential decay signature due to the substantial effect caused by the modal dispersion of optical fibers. In a step-index multimode fiber, the group velocity of the signal pulse is slower if the number of modes is higher, causing the dispersion of the pulse shape due to the different group velocities. The intrinsic approach to resolve modal dispersion is using a graded-index multimode fiber. It uses a relatively higher refractive index at the core and a lower one at the outer region, compensating for the group velocity with the refractive index.

Unfortunately, the market situation does not allow it as a viable solution because of expensive clear fibers. Furthermore, no graded-index scintillating fiber is available commercially. Therefore, we took the software compensation to tackle this issue by assigning faster group velocity for early components of the pulse shape, and a slower one for late components after decay removal.

The group velocity is profiled as a function of $\Delta T$—the time passed from the time of arrival $t_0$. We used the well-understood longitudinal profile of the EM shower from the EGS4 simulation [16], equalizing the relative area of the integrated pulse from $t_0$ to the energy contained from the tail of the shower and the depth $x$, which corresponds to time $t_0 + \Delta T$.

$$\frac{\int_{t_0}^{t_0+\Delta T} f(t)\, dt}{\int_{t_0}^{\infty} f(t)\, dt} = \frac{\int_{x}^{\infty} \frac{dE(x)}{dx}\, dx}{\int_{0}^{\infty} \frac{dE(x)}{dx}\, dx} \tag{7}$$

Then, we can express the group velocity by using $t = t_0 + \Delta T$ and $x$ given by Equation (7).

$$v_{group} = \frac{|\vec{l}| - |\vec{x}|}{t - |\vec{x}|/c} \tag{8}$$

Figure 4 shows the profiled group velocity for scintillation and Cherenkov channel as a function of $\Delta T$. As intended, the group velocity had a slightly lower value than the speed of light within the medium at $\Delta T = 0$, and gradually decreased as $\Delta T$ increases, compensating for the mode increment.

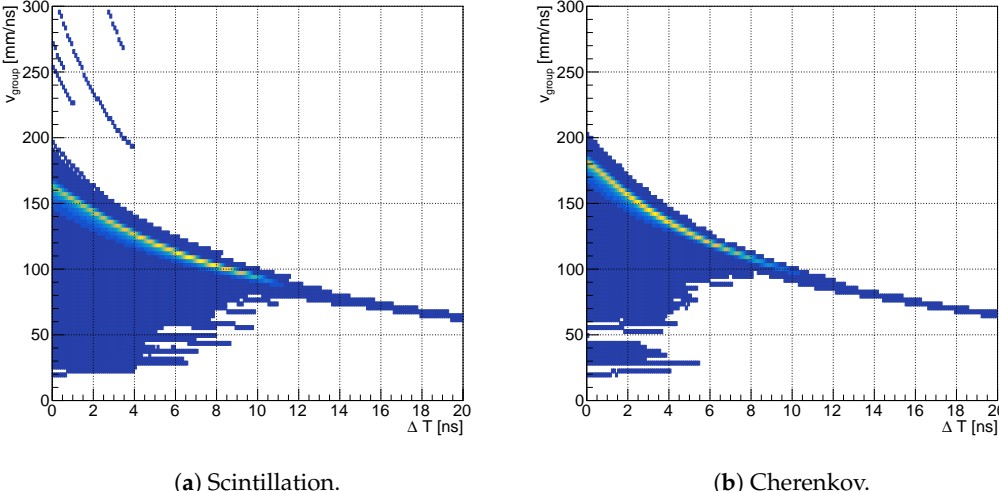

(**a**) Scintillation.

(**b**) Cherenkov.

**Figure 4.** Group velocity for (**a**) scintillation and (**b**) Cherenkov channel as a function of time $\Delta T$ passed since arrival $t_0$, profiled using the signal pulse amplitude per sampling time per SiPM.

### 3.3. Longitudinal Shower Depth, Length, and 3D Shower Shapes

After mitigating modal dispersion within optical fibers, we can describe longitudinal shower shapes using the timing pulses. To test the reliability of reconstructed longitudinal shower shapes, we compared them with the MC truth energy deposits retrieved from GEANT4 steps. We describe the shape with two parameters, depth and length. The depthlike observable is defined as the distance from the tower's front end to the shower maximum, and the lengthlike observable represents the distance between the two positions where the local energy density exceeds 10% of the shower maximum.

Figure 5 shows the depth- and lengthlike observables for 20 GeV electrons and pions. Here, the sampling time of electronics was assumed to be 100 ps, and timing resolution in the order of 10 ps, so that the jitter had a negligible effect. The comparison shows a decent correlation between the reconstructed and MC truth observables. However, the group velocity profiling on the EM shower could not completely correct the dispersion effect and it affected the late arrival photons more strongly. Figure 5a reveals the asymmetrical impact that deviated the head of the reconstructed shower further frontward.

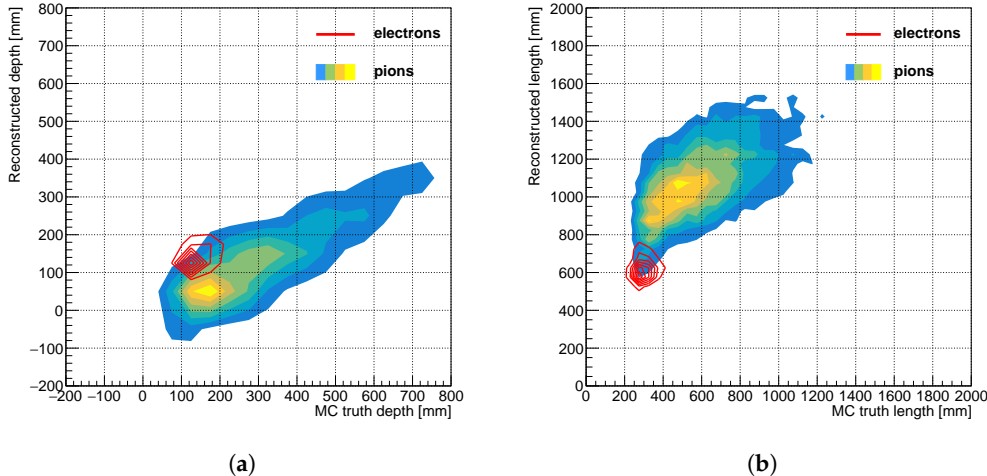

(**a**)

(**b**)

**Figure 5.** Reconstructed vs. MC truth shower (**a**) depthlike and (**b**) lengthlike observables for $10^4$ events of 20 GeV electrons (lined contour) and pions (filled contour), respectively. Contour lines indicate the density of events.

Having descriptions of the longitudinal shower shape, we can illustrate the 3D shower shape reconstructed with the dual-readout calorimeter by mixing it with the lateral shower

shape. Figure 6 renders several event displays of reconstructed 3D shower shape on the left and MC truth on the right.

The reconstructed 3D shower shape of the 20 GeV pion in Figure 6b illustrates typical hadronic shower characteristics that consist of the EM component mainly represented by the Cherenkov hits and the non-EM component by the scintillation hits. The event display shows that the EM component was densely deposited along the center of the shower, while the hadronic component tended to reach deeper and be located away from the center. Furthermore, Figure 6c,d suggest that 3D reconstruction reveals the unique shape of hadronic punch-thru and minimal ionizing particle (MIP), allowing for us to identify them using the shower substructure analysis that was not possible beforehand for particles with arbitrary incident energy.

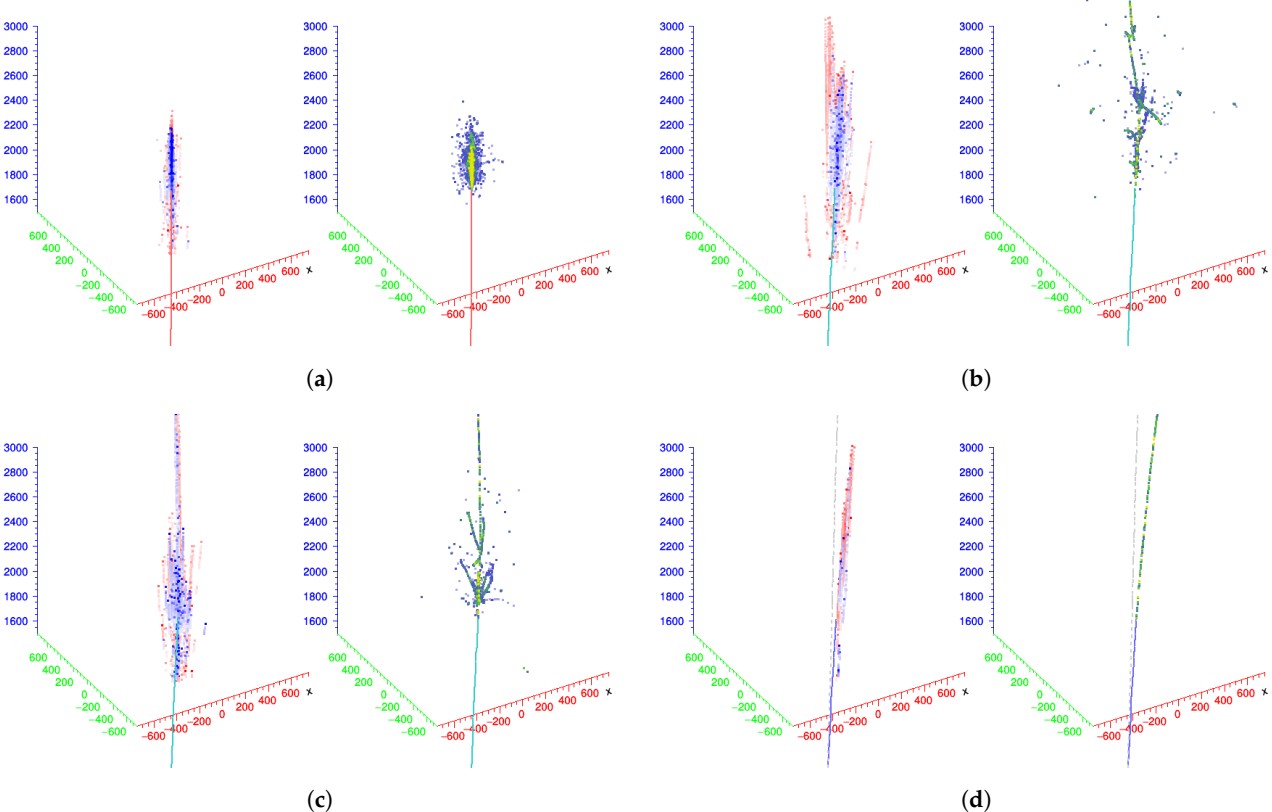

**Figure 6.** (**left**) Reconstructed and (**right**) MC truth 3D shower shapes of 20 GeV particles. For reconstructed cases, red dots represent the 3D hits from the scintillation channel and blue dots from the Cherenkov channel. The color code portrays the relative energy density for the MC truth case. $E_S$, $E_C$, and $E_{DR}$ indicate the reconstructed energy in GeV from the scintillation, Cherenkov channel, and the dual-readout corrected energy. The 2 T magnetic field was applied to all four cases. (**a**) Electron ($E_S = 19.96$, $E_C = 20.84$). (**b**) Pion ($E_S = 17.29$, $E_C = 10.22$, $E_{DR} = 20.04$). (**c**) Punch-thru ($E_S = 13.05$, $E_C = 8.410$, $E_{DR} = 14.85$). (**d**) Muon ($E_S = 1.550$, $E_C = 1.243$).

## 4. 3D Shower Substructure with Density-Based Clustering

We attempted to take advantage of the novel 3D shower shape reconstruction with the dual-readout calorimeter by looking into the properties of its shape. Counting the number of substructures is the simplest way to define the characteristics of a given shower. The DBSCAN algorithm [17] is used to cluster hits from 3D reconstruction, and it has several handy features to cluster shower substructures.

The nature of hadronic shower fluctuation forbids us from knowing how many substructures there are within the shower or where particles head after the scattering process. The DBSCAN allows for us to cluster shower substructures under these circumstances due to its feature that does not require the number of clusters a priori and works on arbitrarily

shaped clusters. Furthermore, it can weigh each point by the 3D hit's energy, which equals the amplitude of the pulse shape after the Fourier transformation at the corresponding time.

The DBSCAN has two input parameters—the maximal distance between the neighboring points within the same cluster (*eps*) and the minimal number of weighted points to create a separate cluster (*minPts*). Considering the lateral granularity of the dual-readout calorimeter (1.5 mm), the *eps* parameter was set to 7.5 mm, and the *minPts* parameter was set to 0.1% of the total number of 3D hits to incorporate as many shower fragments as possible. We equalized each point's weight to the amplitude of the pulse at the corresponding time after the Fourier transformation so that the DBSCAN took the position and energy of 3D hits for the clustering.

However, the DBSCAN did not consider the different lateral and longitudinal accuracy. In the longitudinal direction, it did not have any physical segmentation as in the lateral case. Instead, it relied on the sampling time for determining the depth of each point, where 100 ps of sampling time coincided with a 4 cm bin along the longitudinal axis. Therefore, we scaled the longitudinal axis by a factor of 20 to match the lateral and longitudinal direction accuracy.

Figure 7 depicts the clustered shower substructure for 20 GeV particles. The number of colors in Figure 7b shows that the DBSCAN separately distinguished substructures within the hadronic shower. However, in the attempt to discriminate two photons from the neutral pion decay in Figure 7c, the *eps* parameter had to be reduced to 5 mm. This possibly indicates that the appropriate DBSCAN parameter is a subject of optimization based on the overall size of the shower, since different scales between the radiation length and the nuclear interaction length may require different scopes for substructural clustering.

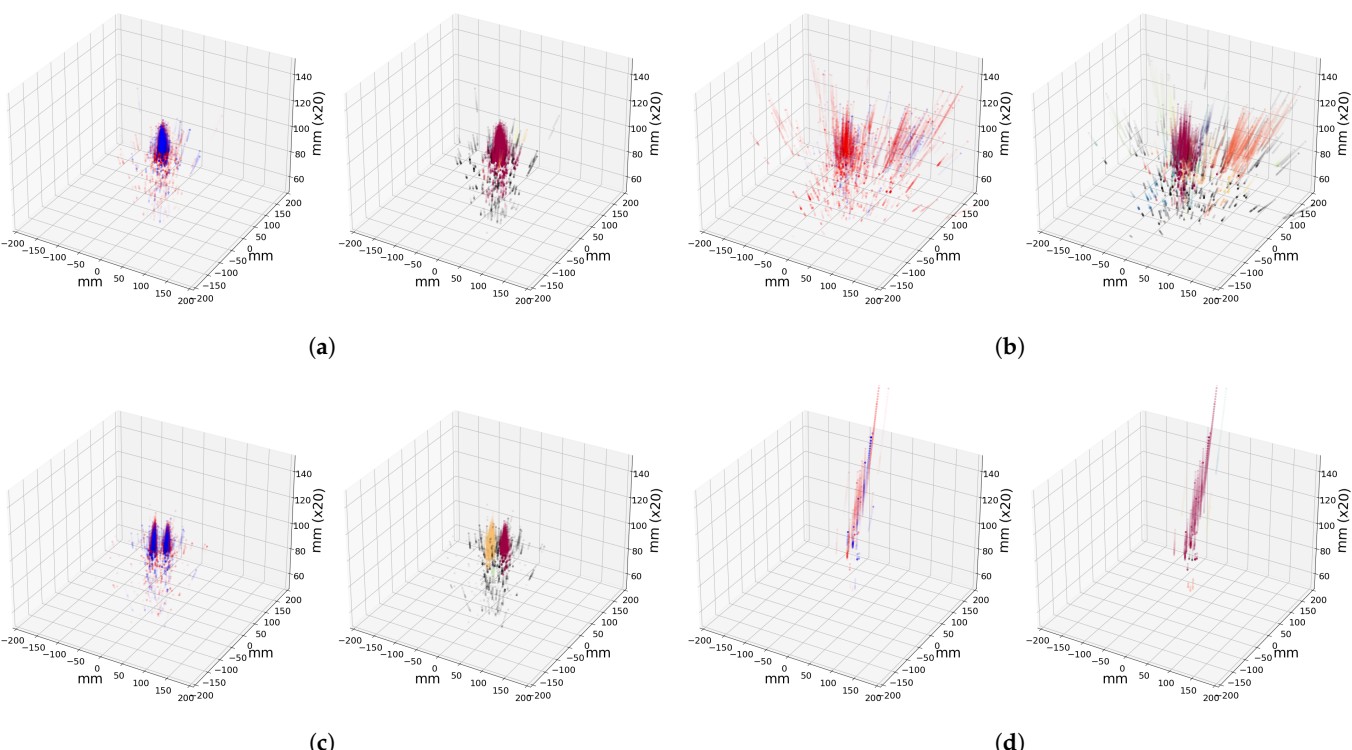

**Figure 7.** (**left**) Reconstructed 3D hits and (**right**) 3D hits after clustering shower substructures. On the left pad of each figure, the red dots represent 3D hits from the scintillation channel and blue dots from the Cherenkov channel. Color codes on the right pad illustrate different substructures clustered by the DBSCAN. The longitudinal (vertical) axis is scaled by 20, where a unit length equals 20 mm. (**a**) Electron. (**b**) Pion. (**c**) Neutral pion. (**d**) Muon.

The estimated number of clusters provides an additional source of information from the C/S variable of the dual-readout calorimeter. Thus, mixing the number of clusters with the C/S can bring insights into the behavior of rare showered particles. For instance, as seen in Figure 8, it allows for us to distinguish hadronic showers not only from EM showers but also showers initiated by particles acting like MIPs, such as punch-thru and muons.

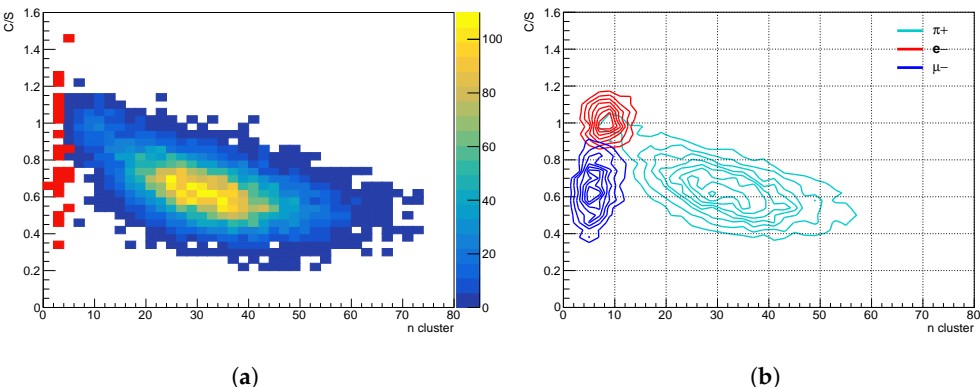

(**a**)                                                                                                         (**b**)

**Figure 8.** The number of clusters vs C/S for (**a**) 20 GeV pions (palette) overlain by the ones with the dual-readout corrected energy less than 5 GeV (red), (**b**) 20 GeV pions (cyan), electrons (red), and muons (blue).

## 5. Summary and Future Developments

Longitudinal and 3D shower shape reconstruction with a dual-readout calorimeter was presented. Despite no physical longitudinal segmentation, the comparison with reconstructed and MC truth shower shapes suggests that exploiting timing information using signal processing may allow for us to reconstruct shower substructures without losing details. Moreover, clustering with DBSCAN reveals that the amount of information contained in the reconstructed shower substructures is substantial to perform basic particle identification by using only the number of clusters mixed with the dual-readout calorimeter's C/S variable.

In this study, we strictly relied on the software-based approach to demonstrate physics and detector response via the simulation and emulation of SiPMs. The simulation study shows that the detector's fine lateral granularity and excellent timing characteristics are essential for reconstructing 3D shower shapes. We plan to perform a beam test at the CERN SPS facility heading towards complete proof of concept by collecting some waveforms of each fiber using Hamamatsu S14160-1310PS SiPMs with a fast decay time and the DRS4 digitizer [18] with 200 ps sampling time.

**Author Contributions:** Conceptualization, S.K.; methodology, S.K.; software, S.K.; validation, S.K.; formal analysis, S.K.; investigation, S.K.; resources, S.K.; data curation, S.K.; writing—original draft preparation, S.K.; writing—review and editing, S.K. and H.Y.; visualization, S.K.; supervision, H.Y. and S.H.; project administration, H.Y.; funding acquisition, H.Y. All authors have read and agreed to the published version of the manuscript.

**Funding:** This work was supported by the Yonsei University Research Fund (Post Doc. Researcher Supporting Program) of 2022 (project no.: 2021-12-0147) and the National Research Foundation of Korea (NRF) grants NRF-2020R1A2C3013540 and NRF-2021K1A3A1A79097711.

**Data Availability Statement:** Not applicable.

**Conflicts of Interest:** The authors declare no conflict of interest.

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
