# Peer review of "Reconstruction of 3D Shower Shape with the Dual-Readout Calorimeter"

_instruments, doi:10.3390/instruments6030039_

Round 1

Reviewer 1 Report

This is in an interesting contribution and I applaud the effort.  However, some modifications are needed.  I won’t dwell on the linguistic problems, but if possible, a native English speaker with good writing skills should review the paper.

The paper draws on lots of different topics: waveform transforms, fiber modal propagation times, clusters, etc. At the end, it is hard to see how all these things come together for a better calorimeter.  Because this information was already presented in the conference, it is fair to include them here, but if this were to be a publication outside the Proceedings, I would request quite a bit more explanation that would certainly expand the content beyond what was presented.

L2: “… outstanding achievements …” This is a value judgement and has no place here.  Perhaps replace this with “applications”?

L4:  “Therefore, there were several efforts to speculate …” Awkward.  How about “There are ongoing efforts investigating the possibility of longitudinal segmentation by timing in fiber calorimeters.”

L5: “The dual-readout calorimeter is one of them.”  What does this mean?  There is no dual-readout fiber calorimeter at the LHC.  Although the idea was investigated by the DREAM Collaboration years ago, it did not find a home in any running experiment.

I suggest the abstract should be completely rewritten in a clear and simple manner to reflect what the reported study involves.  Say something about what you have done.  What is the result?  What’s good?  What’s bad?

L13-L22: The first paragraph is confusing and sounds wrong.  Either rewrite it to be clear about what’s argued or remove it entirely.  It sounds like EM vs HAD segmentation is somehow mixed with the EM fractions…. 

L23: “… [1][2]…” -> [1, 2] also in L42, L43, L45 and other places

L25: Please do not introduce non-English characters into the text.  The correct spelling is “Cherenkov”, not \check{C}

L27:  What’s the purpose of this statement here?:  “… (C/S) makes it not necessary…”  Please explain.  As it stands, it is just wrong.

L57: “…yields…” -> yield

L61-63:  I think I understand the statement that either the peak or the time of arrival is used as a measure of energy deposition along the depth of the calorimeter.  It’s a bit awkward though.  Could you try rephrasing it more clearly?

Figure 3 and Eqs 4-6:  Fourier and other transforms are commonly used in pulse shape analysis to quantify some feature of the pulse shape.  But what is missing here is the punch line:  How is this information used to improve the performance of a calorimeter?

L112: equation misspelled

L128 “… competent …”. What does this mean?

L140: “… sprawls…”. Please rephrase.

It would be nicer to make Figures 6 and 7 larger so that the reader can see the differences and color coding clearly. 

L165: “…(eps)…” ?

What does Figure 9 add to the discussion in the paper?

Author Response

Thank you for the review, we appreciate your time and efforts in providing valuable feedback. We have addressed all of your comments in the revised version. Please find our point-by-point response to your comments and concerns below.

Point 1
Comment:
I suggest the abstract should be completely rewritten in a clear and simple manner to reflect what the reported study involves. Say something about what you have done. What is the result? What's good? What's bad?
Response:
We found that multiple reviewers pointed out that the abstract needs to be improved or reorganized. The abstract has been completely rewritten, and we removed sentences that are not related or presented in the main body. Instead, we briefly introduced the dual-readout calorimeter's design that motivated our study, and what was presented in the article.

Point 2
Comment:
L13-L22: The first paragraph is confusing and sounds wrong. Either rewrite it to be clear about what's argued or remove it entirely. It sounds like EM vs HAD segmentation is somehow mixed with the EM fractions…. 
Response:
The first paragraph has also been rewritten. Instead of confusing mentions on the EM vs Had segmentation, we focused on introducing the difficulty of measuring hadronic showers' energy that motivated the dual-readout concept.

Point 3
Comment:
L27: What's the purpose of this statement here?: "… (C/S) makes it not necessary…" Please explain. As it stands, it is just wrong.
Response:
As your concern, we tried to make it clear that the longitudinally unsegmented design is not a core idea of the dual-readout calorimeter. Instead, we added a sentence in the next paragraph to explain why most contemporary designs of the dual-readout calorimeter have minimal longitudinal segmentation, which becomes the motivation of our study.

Point 4
Comment:
L61-63:  I think I understand the statement that either the peak or the time of arrival is used as a measure of energy deposition along the depth of the calorimeter. It's a bit awkward though. Could you try rephrasing it more clearly?
Response:
Your understanding is correct. We separated conjoined two parts of the sentence by a comma, and rephrased them more clearly.

Point 5
Comment:
Figure 3 and Eqs 4-6:  Fourier and other transforms are commonly used in pulse shape analysis to quantify some feature of the pulse shape. But what is missing here is the punch line:  How is this information used to improve the performance of a calorimeter?
Response:
We replaced Figure 3 with an example of SiPM's emulated signal to make the illustration more useful. Also, We added a paragraph to explain how Fourier analysis with SiPM's signal can contribute to the longitudinal shower shape reconstruction.

Point 6
Comment:
It would be nicer to make Figures 6 and 7 larger so that the reader can see the differences and color coding clearly. 
Response:
We made Figures 6 and 7 a bit larger by reducing whitespaces, but we couldn't enlarge them further due to the limitation with styling. Instead, we rescaled the axis and zoomed in on the plots to make the substructures more visible.

Point 7
Comment:
L165: "…(eps)…"?
Response:
We found that the section lacks information on the DBSCAN's input parameters. We added a paragraph explaining each parameter and how it was set in the context of the calorimeter.

Point 8
Comment:
What does Figure 9 add to the discussion in the paper?
Response:
Indeed Figure 9 is not related to the main body, but it was added to show our hardware efforts to demonstrate our idea. However, due to the significantly increased amount of the manuscript, we decided to remove Figure 9 and briefly mention our hardware efforts in a single sentence.

Lastly, all confusing statements with misused vocabulary, grammatical errors, and LaTeX formatting issues have been corrected, which are pointed out in the following comments:
* L25: Please do not introduce non-English characters into the text. The correct spelling is “Cherenkov”, not \check{C}
* L57: “…yields…” -> yield
* L112: equation misspelled
* L128 “… competent …”. What does this mean?
* L140: “… sprawls…”. Please rephrase.

We thank you again for your valuable feedback and look forward to hearing you on our revised article.

Best regards,
Sanghyun.

Reviewer 2 Report

It is pleasant to read this nice study. I would recommend it for publication after clarifying some aspects of the methodology as suggested in the following.

* Please add the length of the detector in the simulation setups so that readers have a better idea about where the shower locates when it comes to the 3D displace of the shower

* Figure 3 is not discussed in the text. This figure can be useful if a simulated signal is used for illustration and axes are labeled. Comparing (a) with (c), does the "removing the exponential decay signature" method simply removes the tails in these peak time?

* Figure 5: what does the color stand for? What is the third dimension allowing to draw the contour? Please use the same range and binning for x-axis and y-axis so that one can easily compare Reco with MC. In addition, MC and Reco themselves do not have units. MC [mm] -> shower depth in MC [mm]; Reco [mm] -> shower depth in Reco [mm]. Maybe use heatmap? In addition, are the differences only from the smearing effect? If not, please add more explanation. I thought the smearing effect changes the resolution but keeps the nominal values.

* For the clustering method in Section4, please clarify the DBSCAN inputs. What information associated with the 3D shower shape is used in DBSCAN.

* line 167, I find the statement of "orthogonal source" is too strong a statement. It has a clear correlation with the C/S values for hadrons.

Author Response

Thank you for agreeing to review our manuscript, we appreciate your time and efforts in providing valuable feedback. We have addressed all of your comments in the revised version. Please find our point-by-point response to your comments below.

Point 1
Comment:
Please add the length of the detector in the simulation setups so that readers have a better idea about where the shower locates when it comes to the 3D displace of the shower
Response:
Thanks for pointing it out, indeed the length of the detector was missing in the description. We have added the longitudinal dimension to the 'Simulation setups' section.

Point 2
Comment:
Figure 3 is not discussed in the text. This figure can be useful if a simulated signal is used for illustration and axes are labeled. Comparing (a) with (c), does the "removing the exponential decay signature" method simply removes the tails in these peak time?
Response:
We replaced Figure 3 with an example of SiPM's emulated signal to make the illustration more useful. An example of emulated SiPM signal with the new figures shows that the decay removal with Fourier transform may work in a more complicated manner due to the existence of a rise time. Also, we have added a paragraph to explain further how the Fourier transform technique described in the section is helpful in the context of longitudinal shower shape reconstruction.

Point3
Comment:
Figure 5: what does the color stand for? What is the third dimension allowing to draw the contour? Please use the same range and binning for x-axis and y-axis so that one can easily compare Reco with MC. In addition, MC and Reco themselves do not have units. MC [mm] -> shower depth in MC [mm]; Reco [mm] -> shower depth in Reco [mm]. Maybe use heatmap? In addition, are the differences only from the smearing effect? If not, please add more explanation. I thought the smearing effect changes the resolution but keeps the nominal values.
Response:
We have synchronized the x-axis and y-axis, replaced the axis title with proper names of observables, and added a description of the contour. Also, we added a further explanation of the asymmetrical behavior that shifts the nominal value - the dispersion effect is larger for late arriving photons than early arriving photons, hence the head part of the shower is smeared frontward while the tail part of the shower is relatively untouched.

Point 4
Comment:
For the clustering method in Section4, please clarify the DBSCAN inputs. What information associated with the 3D shower shape is used in DBSCAN.
Response:
We added a new paragraph to describe two input parameters of the DBSCAN, in the context of the geometry of the dual-readout calorimeter. The paragraph also explains how the weight information taken by the DBSCAN is associated with the energy in 3D hits.

Point 5
Comment:
line 167, I find the statement of "orthogonal source" is too strong a statement. It has a clear correlation with the C/S values for hadrons.
Response:
We agree that the statement can be too strong, and we replaced it with "additional source".

We thank you again for your valuable feedback and look forward to hearing you on our revised manuscript.

Best regards,
Sanghyun.

Reviewer 3 Report

This paper describes a modern alternative to traditional longitudinally segmented calorimeters through the use of optical fibres, silicon photomultipliers (SiPMs), and software Fourier algorithms to reconstruct particle shower shapes simultaneously for electromagnetic and hadronic objects on the basis of timing information collected via Cherenkov and scintillation processes.  This is an interesting and important area of instrumentation development, and the paper makes a significant contribution to the literature by outlining a simulated implementation of the novel approach of dual-readout calorimetry, the removal of exponential features in the intra-fibre decay signatures, the use of a software compensation method to correct for modal dispersion in the optical fibres, and the reconstruction of particle shower object geometries in 3D.

The topic is introduced well, but would benefit from additional specific contextual information as to the uniqueness of this group's work within the advanced calorimetry community.  For example, it would be beneficial to discuss briefly other competing instrumentation approaches that aim to make use of the particle-flow paradigm to achieve calorimetric measurements in next generation detector designs.

While the main body of the paper is well organized overall, the abstract has two problematic features: (a) it contains information that is not present in the paper body, and (b) it lacks certain important contextual information.  Some modest reorganization is recommended to assist readers.  With regard to (a), the main body of the paper should ideally convey a similar meaning to what is expressed in several sentences of the abstract: the rapid evolution of the field, the "IDEA detector concept", its proposed usefulness for future electron-positron colliders, etc.

With regard to point (b) above, the abstract would very usefully benefit from an implicit definition of the "dual" nature of the readout by explicitly mentioning the terms "electromagnetic" and "hadronic", while also conveying that the readout of these aspects is simultaneous.  As currently written, the abstract could be misunderstood to suggest that the "dual" nature of the readout rather relates to lateral and longitudinal information.

Along the same lines, while the abstract does mention the centrally important timing-information aspect, it does not provide the reader with any statement reminding that the timing readout here obviates the traditional longitudinally segmented calorimeter designs.

Three of the figures have no reference within the body of the paper text: specifically, these are Figures 1, 3, and 9.  Both Equations (1) and (2) have two equations indicated and separated by a comma; I recommend separating both of these arrangements with some significant white space and removing the comma, which could be misunderstood to denote a "prime" symbol.

Specific detailed comments:

L27: "not necessary" -> "unnecessary"

L43: "We reflected" -> "We emulated" (?)

L50: "plugged-in into" -> "plugged into"

Figure 2, caption: "left bottom to the right upper" -> "bottom left to the upper right"

L119: "describe competent" -> "describe complete" (?)

L154: "can weigh each" -> "can weight each"

L166: The fact that the DBSCAN parameter is expected to differ in the electromagnetic versus the hadronic shower-clustering cases seems to warrant a comment, namely that this is a form of tuneable software-based calorimeter compensation.

Author Response

Thank you for agreeing to review our manuscript, we appreciate your time and efforts in providing valuable feedback. We have addressed most your comments in the revised version. Please find our point-by-point response to your comments below.

Point 1

Comment:

The topic is introduced well, but would benefit from additional specific contextual information as to the uniqueness of this group's work within the advanced calorimetry community. For example, it would be beneficial to discuss briefly other competing instrumentation approaches that aim to make use of the particle-flow paradigm to achieve calorimetric measurements in next generation detector designs.

Response:

We have added some contextual information about the software compensation technique of particle flow calorimeters, which provides an alternative method to improve the energy measurement of hadronic showers. Also, it emphasizes the importance of longitudinal shower shape reconstruction, which contributes to the motivation of our study.

Point 2-a

Comment:

While the main body of the paper is well organized overall, the abstract has two problematic features: (a) it contains information that is not present in the paper body, and (b) it lacks certain important contextual information. Some modest reorganization is recommended to assist readers. With regard to (a), the main body of the paper should ideally convey a similar meaning to what is expressed in several sentences of the abstract: the rapid evolution of the field, the "IDEA detector concept", its proposed usefulness for future electron-positron colliders, etc.

Response:

We found that multiple reviewers pointed out that the abstract needs to be improved or reorganized. The abstract has been completely rewritten, and we removed sentences that are not related or presented in the main body of the manuscript. Instead, we decided to briefly introduce the dual-readout calorimeter's design, which connects to the motivation of our study.

Point 2-b

Comment:

With regard to point (b) above, the abstract would very usefully benefit from an implicit definition of the "dual" nature of the readout by explicitly mentioning the terms "electromagnetic" and "hadronic", while also conveying that the readout of these aspects is simultaneous. As currently written, the abstract could be misunderstood to suggest that the "dual" nature of the readout rather relates to lateral and longitudinal information.

Along the same lines, while the abstract does mention the centrally important timing-information aspect, it does not provide the reader with any statement reminding that the timing readout here obviates the traditional longitudinally segmented calorimeter designs.

Response:

In the reorganized abstract, we believe that the term 'dual' should be clear to the readers given by the short introduction on the dual-readout calorimeter with explicit statements about Cherenkov and scintillation channel as well as EM and hadronic components. We also tried to state the association between the 3D shower shape from the timing and C/S information from the dual-readout discussed in the main body.

Point 3

Comment:

Three of the figures have no reference within the body of the paper text: specifically, these are Figures 1, 3, and 9. Both Equations (1) and (2) have two equations indicated and separated by a comma; I recommend separating both of these arrangements with some significant white space and removing the comma, which could be misunderstood to denote a "prime" symbol.

Response:

We added further explanations regarding Figures 1 and 3 in the main body of the text. Figure 9 has been removed due to another reviewer's comment (little relation to the main body), and a significant increase in the amount of the manuscript after revision. Both equations (1) and (2) were modified to be separated with white space instead of the comma.

Point 4

Comment:

L166: The fact that the DBSCAN parameter is expected to differ in the electromagnetic versus the hadronic shower-clustering cases seems to warrant a comment, namely that this is a form of tuneable software-based calorimeter compensation.

Response:

If I understand the term 'software-based calorimeter compensation' correctly (what I understood is the context of 'software compensation' in particle flow calorimetry), it seems that the statement in the manuscript sounds misleading. We rephrased and extended the sentence by explicitly stating the word 'optimization' and 'size of the shower'. In the study, the role of DBSCAN is limited to spatial clustering, but not to the compensation of different responses between the EM and hadronic components.

Lastly, we addressed the following grammatical or misuse of vocabulary (unless completely rewritten):

  • L27: "not necessary" -> "unnecessary" (the sentence is rewritten at the request of another reviewer)
  • L43: "We reflected" -> "We emulated" (?)
  • L50: "plugged-in into" -> "plugged into"
  • Figure 2, caption: "left bottom to the right upper" -> "bottom left to the upper right"
  • L119: "describe competent" -> "describe complete" (?) (rewritten in a more simple manner)
  • L154: "can weigh each" -> "can weight each"

We thank you again for your valuable feedback and look forward to hearing you on our revised manuscript.

Best regards,

Sanghyun.

Round 2

Reviewer 1 Report

A few corrections from my re-reading of the second version:

L48 Why “minimize longitudinal segmentation”. Not clear what is meant by this. 

L95 $v$

L155 Equation 7.

L175 it affect late arriving photons more strongly ?  Not “heavier”

I think the manuscript will be ready to appear in the Proceedings once a careful editing for linguistic issues is carried out.  

Author Response

Thank you for the second review, and we're glad to see your response to our revised manuscript. We have corrected the linguistic & formatting issues that you pointed out:

  • L48 Why "minimize longitudinal segmentation". Not clear what is meant by this
    ("minimize longitudinal segmentation" has been changed to "have no longitudinal segmentation" to be more explicit)
  • L95 $v$
  • L155 Equation 7.
  • L175 it affect late arriving photons more strongly? Not "heavier"

Moreover, we will carry out further proofreading carefully before submitting the second revision. Thank you again for your valuable feedback.

Best,
Sanghyun.